# Unveiling the role of DHIS2 in enhancing data quality and accessibility in primary healthcare facilities: Evidence from Ethiopia

**Taddese Alemu Zerfu** [1] *, **Moges Asressie**[2], **Zenebu Begna**[1,3], **Tigist Habtamu**[4], **Netsanet Werkneh**[5], **Tariku Nigatu**[1], **Meskerem Jisso Ibido**[6], **Addisalem Genta**[1,3]

**1** EMIRTA Institute of Research, Training and Development, Addis Ababa, Ethiopia, **2** Planning, M & E Office, Amhara Regional Health Bureau, Bahir-Dar, Ethiopia, **3** Department of Public Health, Ambo University, Ambo, Ethiopia, **4** Addis Ababa University (AAU), Addis Ababa, Ethiopia, **5** Jimma University, Jimma, Ethiopia, **6** Hawassa University, Hawassa, Ethiopia

* tadalzerfu@gmail.com

## Abstract

**Data Availability Statement:** The data underlying the results presented in this study are freely and openly available. Data can be accessed through the

### Background

The implementation of DHIS2 in healthcare systems has transformed data management practices worldwide. However, its specific impact on data quality, availability, and performance in Primary Health Unit (PHU) facilities in Ethiopia remains underexplored. Therefore, we investigated the contribution of DHIS2 to enhancing data quality, availability, and performance within PHU facilities in Ethiopia.

### Methods

We employed qualitative methods, specifically Key Informant Interviews (KIIs) and Focus Group Discussions (FGDs), to gather insights from stakeholders, including healthcare providers and administrators at PHCUs across Ethiopia. Convenience sampling was used for FGDs, while purposive sampling targeted key informants with relevant expertise. Data were systematically analysed thematically, identifying patterns and themes related to DHIS2's impact on data management within PHUs. This approach offered a comprehensive understanding of the system's effectiveness and the factors influencing its implementation, highlighting both successes and challenges in integrating DHIS2 into healthcare practices.

### Findings

Participants from various regions reported significant enhancements in the timeliness, completeness, accuracy, and accessibility of health data following the implementation of DHIS2. While some concerns were raised regarding variations in reporting intervals, the consensus indicated marked improvements in data management processes. DHIS2 standardized data collection methods, enabling healthcare providers to input and access data in real-time. This advancement fostered greater accountability and transparency within the healthcare system. Additionally, unexpected benefits arose, including increased digital literacy among staff, equipping them with necessary skills for effective data management, and the creation

public repository Dryad at https://doi.org/10.5061/dryad.k98sf7mh2. Additionally, interested parties may request the data by contacting the corresponding author or the EMIRTA Institute of Research, Training, and Development at info@emirta.org.

**Funding:** Funding for this project was obtained from WHO Alliance for Health Policy and Systems Research. The funders had no role in study design, data collection and analysis, decision to publish, or preparation of the manuscript.

**Competing interests:** The authors have declared that no competing interests exist.

of job opportunities, particularly for youth. Ultimately, DHIS2 emerged as a pivotal tool for enhancing data quality and promoting health service equity across Ethiopia.

## Conclusion

DHIS2 has significantly improved data quality and accessibility in Ethiopia, enhancing healthcare management and accountability across facilities. Healthcare providers should continue to leverage its robust features and prioritize ongoing staff training to improve digital literacy and data management skills. Establishing consistent reporting practices and regular audits will further maintain data integrity and foster a culture of accountability within the healthcare system.

## Introduction

In recent years, the advancement of digital health information systems has reshaped the landscape of healthcare management worldwide [1, 2]. Among these innovations, the District Health Information System 2 (DHIS2) has emerged as a linchpin in enabling the collection, analysis, and utilization of health data, particularly in resource-constrained settings like Ethiopia [3, 4]. Access to reliable and timely health information is crucial for informed decision-making and efficient resource allocation, making the adoption of DHIS2 a game-changer in primary healthcare (PHC) data management [5, 6].

Globally, DHIS2 has been implemented in various low and middle-income countries (LMIC), demonstrating its potential to improve health outcomes [7]. For instance, in countries with limited resources and infrastructure, DHIS2 has facilitated better data collection practices, leading to improved monitoring of health indicators [8]. The system's standardized approach to data management promotes consistency, enhances the reliability of health information, and supports data-driven policy decisions.

The implementation of DHIS2 in Ethiopia was a strategic response to the limitations of traditional paper-based health systems [9, 10]. By embracing digital platforms, Ethiopia has effectively addressed issues including data collection inefficiencies, quality discrepancies, and limited accessibility to health information [11, 12]. DHIS2's user-friendly interface equips healthcare workers with intuitive tools for real-time data collection, analysis, and reporting, revolutionizing healthcare management practices. This shift not only streamlines administrative tasks but also fosters a culture of data-driven decision-making within the healthcare sector [13, 14].

Moreover, the adoption of DHIS2signifies a transformative leap towards modernizing healthcare information systems [15, 16]. It empowers healthcare workers to use technology to enhance patient care and optimize resource allocation. Real-time data analysis and reporting facilitated by DHIS2 improve the efficiency and effectiveness of healthcare delivery, ultimately leading to improved health outcomes for Ethiopian communities [3, 17].

The DHIS2 system has played a crucial role in enhancing the quality, availability, and performance of data within primary healthcare units (PHUs) in Ethiopia [4, 18]. By providing a robust platform for real-time data collection and analysis, DHIS2 has transformed information management in healthcare facilities, leading to improved decision-making and resource allocation. It effectively addresses critical challenges such as data inaccuracies and delays inherent in traditional paper-based methods [18].

Despite these recognized benefits, significant gaps persist in understanding DHIS2's adoption and implementation in low-resource settings. Comprehensive research is lacking on the contextual factors influencing successful integration, including infrastructural limitations, training needs, and cultural attitudes toward digital tools. Additionally, the perspectives of frontline healthcare workers, who interact with the system daily, are often overlooked, raising questions about usability and practicality. Moreover, the long-term impact of DHIS2 on health outcomes and data quality remains inadequately explored, particularly regarding disparities in service delivery across different regions.

This study aims to bridge these gaps by thoroughly examining DHIS2's contributions in Ethiopia. By highlighting its transformative potential, the manuscript seeks to inform evidence-based policymaking and support sustainable development efforts in Ethiopia and other regions facing similar healthcare challenges.

## Methods

### Study design and period

This qualitative study was conducted over a specified period in five strategically selected regions of Ethiopia, including the capital city administration of Addis Ababa and four other regions. The study employed an exploratory qualitative approach, employing both key informant in-depth interviews (KII) and focus group discussions (FGD) to explore the intricacies of DHIS2 utilization in the context of primary healthcare (PHC). Data collection and analysis were conducted over six months, from October to March 2024, ensuring thorough coverage and depth in the findings.

The KII respondents included health managers of primary healthcare institutions such as district hospitals, health centres, and health posts. Additionally, health service leaders/administrators such as heads of district health offices, zonal health departments, regional health bureaus (RHBs), and the federal ministry of health (MOH) were also included in the interviews. The FGDs were conducted with performance management team leaders of district health facilities and health managers from selected districts.

### Study area

The study was primarily conducted at primary health care (PHC) institutions in Ethiopia. These institutions include district hospitals, health centres, health posts, and district health offices, as defined by the Ministry of Health (MOH). The selected PHCUs for the study were chosen from regions representing four different agroecology and economic areas/zones of the country. Specifically, two agrarian regions (Amhara and Oromia), a cash crop producing region (Sidama), a pastoralist region (Somali), and an urban area were included.

Qualitative data was collected through interviews with health managers at zonal health departments, RHBs, and the MOH. Furthermore, interviews or discussions were conducted with performance monitoring team (PMT) members and HIS focal persons at their respective health facilities.

### Sample size and sampling technique

The determination of sample size was guided by the principle of information saturation, ensuring that sufficient data were collected to achieve thematic saturation across diverse subgroups. A combination of key informant interviews and focus group discussions enabled the exploration of multiple perspectives and experiences related to DHIS2 utilization.

Purposive sampling techniques were employed to select participants based on their roles within primary healthcare institutions, level of training, and professional experience. This method ensured the inclusion of a diverse array of stakeholders, from health managers at district hospitals to performance monitoring team leaders at health facilities. In total, we conducted 30 KIIs and 16 FGDs. Most focus groups consisted of 8 participants, although a few included between 7 and 9 participants, resulting in approximately 132 participants across the 16 FGDs.

## Sampling procedure

Convenience and purposive sampling strategies were utilized to identify and recruit participants for focus group discussions and key informant interviews, respectively. Participants were selected from primary healthcare units (PHUs) and health administration offices spanning federal to district levels, ensuring representation from various tiers of the healthcare system. study participants were distributed across the four major livelihood systems (agrarian, cash-crop, pastoralist and urban) to produce a robust and geographically triangulated result.

## Data collection tools and procedures

Structured and unstructured interview guides were developed in English and subsequently translated into local languages, including Amharic, Afan Oromo, and Somali, to facilitate effective communication with participants. A team of trained data collectors, comprising individuals with backgrounds in health, information technology, and behavioural sciences, were deployed to conduct interviews and focus group discussions. Prior to data collection, all team members underwent intensive training to ensure proficiency in administering questionnaires, capturing responses, and maintaining ethical standards.

## Outcomes and measurements

The qualitative study aimed to explore key dimensions of DHIS2 implementation, including data availability, accessibility, accountability, and utilization within the PHC context. These outcomes were assessed through in-depth interviews and focus group discussions, allowing for a nuanced understanding of the multifaceted impact of DHIS2 on healthcare delivery and decision-making processes.

## Data analysis

The data analysis process seamlessly commenced alongside data collection in the field, with verbatim transcripts promptly translated into English for meticulous examination. Employing thematic analysis, a rich tapestry of recurring patterns, themes, and insights emerged from the qualitative dataset. ATLAS.ti software emerged as a crucial ally, facilitating the organization, coding, and interpretation of qualitative data, ensuring both methodological rigor and transparency throughout the analytical journey.

Within the thematic analysis, a nuanced exploration unveiled several pivotal themes and their corresponding subthemes. Anchored under the overarching theme of the perceived impact of DHIS2 on healthcare services' data quality, the timeliness of routine HMIS data took center stage, highlighting the system's notable influence.

Another substantial sub-theme that emerged was the "Effect on Data Completeness," capturing community members' perceptions of the comprehensiveness and inclusivity of health data captured through DHIS2. Subthemes within this realm ranged from the degree of pertinent health indicators captured to the integration of marginalized populations in health data reporting.

Moreover, the sub-theme of "Effect on Data Accuracy" delved into community members' perceptions of the reliability and precision of health data collected through DHIS2, shedding light on the system's impact on data integrity.

Additionally, participants underscored the significance of the "Effect on Reduction of Discrepancies Between Reports," highlighting DHIS2's role in harmonizing data across various health facilities, resolving disparities, and fostering consistency in health data reporting.

Under the sub-thematic lens of "Content (Indicator) Completeness," community members emphasized the imperative of capturing a comprehensive array of health indicators within DHIS2 to drive evidence-based decision-making and program planning.

Furthermore, other notable themes emerged, including the Perceived Impact of DHIS2 on Data Availability and Accessibility, its effect on Data PHCU Accountability, the influence of DHIS2 on Equity of data quality and health services, as well as the serendipitous Unexpected Benefits unearthed through its implementation.

## Ethical considerations

Ethical clearance was secured from IRBs of Public Health Institutes in all five regions and Addis Ababa city administration. Permissions and administrative support were obtained from the federal Ministry of Health and regional health bureaus. All participants provided verbal informed consent, emphasizing voluntary participation, confidentiality, and the right to withdraw without repercussions. Formal documentation was promptly generated in the form of confirmation emails and letters from each IRB. Stringent ethical standards were followed to protect participants' rights and well-being throughout the study.

## Results

### Perceived impact of DHIS2 on data quality at PHUs in Ethiopia

Overall, participants in the qualitative study unanimously agree that DHIS2 has significantly improved the quality of data. The study included participants from all regions and employed methods such as Key Informant Interviews and Focus Group Discussions. Although there may be some differences in certain areas, there is a consensus across regions about the positive impact of DHIS2 on key dimensions of data quality, such as timeliness, availability, accessibility, and accuracy of health and nutrition data. The following paragraphs provide a detailed discussion of each of these components, explaining the improvements observed in timeliness, availability, accessibility, and accuracy, all thanks to DHIS2.

### Effect of DHIS2 on timeliness of routine HMIS

The finding indicates a strong consensus among respondents from all five regions regarding the overall improvement in data quality dimensions compared to pre-DHIS2 periods. Specifically, there is notable agreement on the substantial enhancement in timeliness of data. A PHCU head from one region emphasized the stark improvement in timeliness following DHIS2 implementation, highlighting a significant shift from previous limitations in monitoring data timeliness to clear enhancements in data management reports, saying:

> "*Before DHIS2 came into play in our district, we struggled with a glaring gap in keeping tabs on data timeliness. But once we implemented it, the transformation was palpable–our data management reports now boast a level of timeliness that was previously unimaginable.*"

PHCU head in one of the regions

The consensus among many PHCU leaders, ICT technicians, and other participants is that the implementation of DHIS2 has significantly improved data management processes. This improvement is attributed to the introduction of specific timeframes for data entry and completion, ensuring the timely and accurate input of data. Moreover, DHIS2 standardizes data collection formats, which enhances consistency and overall quality throughout the system.

> "*The introduction of DHIS2 not only streamlines data entry and completion with its predefined timeframes but also assures the validation, timeliness, and accuracy of the entered data. Embracing real-time data entry further enhances data quality. Moreover, DHIS2's uniform data collection formats guarantee consistency and comparability across the entire system, fostering reliable analysis and decision-making.*"

<div align="right">ICT head from an urban PHCU facility</div>

However, in one of the regions, there was a prevailing sentiment that although there has been improvement in data completeness, the timelines within DHIS2 have not changed. This is primarily due to the strict deadline for submitting reports, which is currently set for the 22nd of every month.

> "*In our country, health facilities must submit reports by day 21, with Health Posts sending data to health centers by day 23. Health centers finalize reports by day 26. Notably, some regions submit reports as early as day five, highlighting improved efficiency.*"

<div align="right">Head of a PHCU in one of the districts</div>

In alternative regions, participants expressed apprehension regarding variations in reporting intervals, disparities among individual reports, and discrepancies encountered when soliciting identical reports for the same timeframe. Although these concerns persist to some extent, there has been discernible progress. One FGD participant remarked:

> "*As previously mentioned, disparities in reporting times and inconsistencies among reports from different individuals were observed. Even when requesting identical reports from the same timeframe, disparities persisted. It is noteworthy that while these challenges have not been entirely eradicated, there has been a degree of improvement.*"

## Effect of DHIS2 on data completeness among the PHUs

Participants in the study have emphasized the beneficial influence of DHIS2 in improving data completeness within PHCUs and along the reporting hierarchy. They substantiate their claims by pointing out that reports generated through DHIS2 are thorough and inclusive, crediting this to the system's inherent mechanisms for ensuring completeness.

> "*In our previous reliance on Excel sheets within the former HMIS, error detection in reporting was conducted manually. However, with DHIS2, if figures are not inputted into the 'by-marginal age' classification, the system triggers a red alert, prompting data rectification.*"

<div align="right">Head of a PHCU in Addis Ababa</div>

This notion was further supported by a Health Information Technician (HIT) from one of the PHCUs.

"*Before implementing DHIS2 software for reporting and managing our routine health facility data, our reporting lacked completeness, making it difficult to assess the quality of information. Moreover, data indicators and elements were typically aggregated and transmitted solely as figures. Now, we have access to both indicators and data elements.*"

On the contrary, regarding potential errors in routine health facility reports, they attribute them to limitations or mistakes in recording and record-keeping practices, rather than shortcomings in the DHIS2 software.

"*Often, when we conduct quality checks, the errors we encounter originate from the registers. Therefore, I cannot attribute them solely to issues with Health Information Technicians (HITs)...*"

Head of a district PHC health office from one of the regions.

They also emphasize the significance of establishing and reviewing routine health facility data by the Program Management Team (PMT) before forwarding it to the next higher level.

"*Most of the time, you'll find accurate data on DHIS2 because errors are corrected during PMT and case team meetings,*" noted the Head of a PHCU from one of the regions.

Some PHCU leaders and HIT professionals also argue that DHIS2 achieves completeness not only by reporting expected data but also by enabling the inclusion of a wide range of comprehensive indicators.

"*I firmly believe that DHIS2 has revolutionized our programs. In the past, the ANC program was overlooked in DHIS2, but now it is seamlessly integrated, allowing us to effortlessly track the number of pregnant mothers receiving services and ANC visits. This inclusion has catalysed a positive shift in reducing maternal mortality rates.*"

*FGD* discussant from one of the regions

## Effect on data accuracy

Study participants unanimously concurred that the implementation of DHIS2 for routine data management and reporting has led to a significant improvement in data accuracy, facilitating seamless decision-making processes.

"*DHIS2 advances beyond traditional HMIS methods, which rely on manual processes. Its streamlined approach enhances accuracy and speeds up data entry and management, making it a more efficient and preferred choice for many healthcare organizations.*"

Head of a PHCU from one of the regions

While some challenges persist, such as sporadic inaccuracies in reports, DHIS2 provides convenient access to precise data for prompt decision-making.

"*During data entry, multiple cross-checks are conducted. Initially, accuracy is verified, followed by ensuring completeness and adherence to required timelines. Additionally, the system conducts its own checks, including validation of components.*"

HIT from one of the regions

They also indicated that DHIS2 has enabled corrections and enhancements to inaccurate data, often addressed during PMT and case team meetings, thereby bolstering the credibility of data within the DHIS2 system.

"...In my assessment, while there are areas needing improvement to enhance decision-making, our progress thus far is commendable given our early stages. A notable incident involved ART data, crucial for funding medication. A 50% discrepancy between our report and the actual client numbers prompted urgent data cleaning. Addressing such discrepancies promptly is essential to ensure all clients receive their medication as needed."

<div align="right">HIT from one of the regions</div>

"... Precise data is often encountered within DHIS2, as discrepancies are typically resolved during Program Management Team (PMT) and case team gatherings.

<div align="right">PMT member FGD discussant from one of the regions</div>

## Effect on reduction of discrepancies between reports

Across various regions, participants perceive DHIS2 as a key factor in notably mitigating data discrepancies in reports. They view DHIS2 as a robust instrument for gathering, validating, and analysing health data in Ethiopia, thereby fostering harmonization, and streamlining within the process. Consequently, this culminates in enhanced data quality, facilitating informed decision-making and policy formulation.

"... when contrasting the manual process with the current DHIS2 system, noticeable disparities emerge. Previously, reports often exhibited temporal inconsistencies, where data from the same period varied between iterations. Furthermore, discrepancies in recording and reporting were commonplace."

<div align="right">A regional HIT expert</div>

## Content (indicator) completeness

Study participants assert that DHIS2, incorporating an array of new tools, indicators, and service categories, empowers users to generate diverse datasets and outcomes. This expanded functionality significantly enhances the capacity for comprehensive health monitoring and analysis.

"In its initial stages, the HIMS lacked several crucial program and service indicators. With the adoption of DHIS2, however, there has been a gradual expansion in the indicator repertoire. Starting at 108 indicators, the system progressively grew to 122, then 131, and now encompasses approximately 171 indicators. This extensive range now comprehensively covers various program metrics, reflecting the system's evolution and adaptability"

<div align="right">HIT expert from a region</div>

Likewise, a district head from one of the Primary Health Units (PHUs) underscored the significance of DHIS2 in mitigating data disparities across reports, stating:

"The system now includes 188 indicators, up from 131, enabling a comprehensive evaluation of health status. With DHIS2, our service portfolio has significantly expanded, providing tools that can effectively generate valuable data and outcomes."

<div align="right">District PHCU head from a region</div>

## Perceived impact of DHIS2 On data availability and accessibility

Ever since its inception, DHIS2 has wielded a profound influence on the accessibility of data in numerous low- and middle-income countries (LMICs), heralding a new era of centralized data management and real-time access in the realm of public health. Through its intuitive interfaces, customizable reporting tools, and facilitation of collaborative efforts among stakeholders, DHIS2 stands as a beacon of progress, significantly bolstering health outcomes and fortifying healthcare systems across LMICs.

In Ethiopia, the consensus among nearly all participants in this study resonates with the transformative impact of DHIS2 in breaking down barriers to data access, transcending geographical and temporal constraints. For instance, a PHCU director from one of the regions grappling with data accessibility issues eloquently articulated the pivotal role of DHIS2, stating:

> "*DHIS2 revolutionizes data access, liberating us from HMIS constraints. Previously, accessing data was restricted by time and place. Now, with DHIS2, data is at our fingertips anytime, anywhere, contingent upon stable connectivity. It's a transformative leap in accessibility and efficiency.*"

> Head of a PHCU in one the regions

Another key informant from the region echoed this sentiment, affirming,

> "*The web-based nature of DHIS2 offers a distinct advantage: accessibility from any location with just a username and password. Previously, with paper-based reporting, accessing data beyond the facility was cumbersome. DHIS2 eliminates these barriers, providing on-demand data access, anytime, anywhere.*"

Study participants emphasized the importance of accessing HIMI data via DHIS2 even on weekdays when specific staff members, like HITs, might be unavailable but the data remains crucial for decision-making. A seasoned HIT from Addis Ababa elaborated on this need for continuous access to data for informed decision-making.

> "*On Saturdays and Sundays, we are off duty, but we're still committed to assisting managers remotely if needed. Providing essential information during these days is our priority. Additionally, with an official letter, anyone can access the data, ensuring equitable availability.*"

> Experienced HIT from Addis Ababa

Additional participants fervently highlighted DHIS2's role in ensuring offline data accessibility. They marvelled at its capacity to facilitate seamless data collection, entry, analysis, and decision-making in areas with sporadic internet connectivity. DHIS2's potential to sustain operations despite limited or unreliable online access left a profound impression on them.

> "*In our region, numerous health facilities lack internet, computers, or reliable power for DHIS2 operation. Offline data access has emerged as a vital solution, enabling facilities with computer and electricity access to input data offline. Subsequently, data is transferred via flash drive or external disk to an internet-enabled location, ensuring accessible data retrieval whenever necessary.*"

> Regional expert working on HMIS and data revolution

Furthermore, participants underscored DHIS2's profound impact on health data storage and accessibility. Its centralized database, robust backup and recovery systems, user access control, role-based access, and web-based interface ensure the secure, confidential, and user-friendly storage and retrieval of health data. DHIS2's array of features guarantees long-term data accessibility and usability within the healthcare system, facilitating multi-person access and promoting data security and confidentiality. DHIS2 served as a centralized database for storing health data, ensuring that all information is stored in one secure location. This centralized approach facilitated long-term storage by preventing data fragmentation and loss.

"*As a HIT deeply engaged with health data management, I've witnessed first-hand the transformative power of DHIS2's centralized database. It acts as a secure repository, consolidating all health data into one centralized hub. This approach not only safeguards against fragmentation and loss but also ensures long-term storage, enhancing data integrity and accessibility.*"

Regional HIT technician

DHIS2 incorporates robust data backup and recovery mechanisms to safeguard against data loss. Regular backups were performed to ensure that data is protected and can be restored in the event of system failures or disasters.

DHIS2 allows administrators to set user access permissions, controlling who can view, edit, or delete specific data. This ensured that only authorized personnel have access to sensitive health information, promoting data security and confidentiality.

DHIS2 supported role-based access control, allowing different users to have access to specific data based on their roles and responsibilities within the health system. This ensured that relevant stakeholders, including healthcare providers, managers, and policymakers, accessed the data they need to perform their duties effectively.

"*As head of a Primary Healthcare Unit, DHIS2's role-based access control has been instrumental. Its tailors' data access to users' roles, ensuring our healthcare providers, managers, and policymakers access the precise information required for their responsibilities. This targeted access enhances efficiency and empowers effective decision-making within our health system.*"

Head of a Primary Healthcare Unit from a city

DHIS2 is accessible through a web-based interface, allowing authorized users to access health data from any location with internet connectivity. This promotes multi-person accessibility by enabling healthcare teams and stakeholders to access data remotely, facilitating collaboration and decision-making.

Participants in the study have articulated DHIS2's provision of expansive and user-friendly data management capabilities. Authorized individuals, spanning staff members, managers, and stakeholders, can seamlessly access, generate, and utilize healthcare data. DHIS2's open-source architecture guarantees widespread accessibility, while robust permissions and password controls safeguard data integrity. Moreover, DHIS2 facilitates programmatic access to healthcare data, streamlining data management and analysis within the healthcare ecosystem.

"*With permission, anyone can access data through DHIS2, an open-source platform that allows universal usage. DHIS2's being an open-source platform, data accessibility transcends boundaries. Permission grants universal usage, empowering stakeholders to harness data responsibly. DHIS2 offers unparalleled data availability, surpassing HMIS. Authorized*

*individuals can access it from anywhere, facilitating informed decision-making. Data analysis usernames further empower staff and managers to glean insights while ensuring data integrity."*

District PHCU manager from one of the regions

## Perceived impact of DHIS2 On Data PHCU accountability

DHIS2 is credited with bolstering accountability on various fronts, including error detection and correction, transparency enhancement, and fostering ownership and responsibility. Study participants highlighted DHIS2's role in accountability, noting its adeptness in identifying and rectifying mistakes swiftly through robust validation features. Furthermore, DHIS2's audit trail capability ensures transparent tracking of data activities, instilling confidence in data integrity and minimizing errors. These accountability measures not only enhance data quality but also cultivate trust in health information systems.

*". . .For instance, when editing data in DHIS2, the platform records the time and location of the edit, enhancing accountability by providing clear visibility into data modifications."*

HIT from a PHCU

DHIS2 is lauded for its role in enhancing accountability through improved transparency, achieved through various mechanisms. Central to this is its provision of a centralized data platform, facilitating seamless oversight and management. Authorized users benefit from unified access to data, promoting transparency as stakeholders monitor health indicators and outcomes in real-time. DHIS2's additional features, including data validation rules and audit trails, further bolster transparency by identifying errors and unauthorized alterations, ensuring data precision and integrity. Ultimately, this fosters informed decision-making, as decision-makers rely on reliable, transparent information for evidence-based planning.

*". . . We strictly adhere to protocol; information is only released upon presentation of an official letter from the health bureau or sub-cities. It's important to note that the data I possess may vary from incoming information."*

PMT member FGD discussant from Addis Ababa

Furthermore, DHIS2 is noted for its facilitation of data sharing and reporting across different tiers of the healthcare system, spanning from local facilities to national entities and international partners. This transparency fosters collaboration among stakeholders at all levels, facilitating the exchange of information and fostering mutual accountability in achieving health objectives. DHIS2's accountability mechanisms include enhancing data management transparency, ensuring data accuracy and integrity, and facilitating collaboration and information exchange throughout the healthcare continuum.

DHIS2 empowers users within the healthcare sector by fostering ownership and responsibility in data management. Through decentralized data entry and management, local health facilities and personnel take charge of their data, promoting accountability and informed decision-making at the facility level. Role-based access control ensures transparency by assigning specific responsibilities to users. Administrators designate roles like data entry clerks and program coordinators, ensuring everyone understands and fulfils their duties. This approach not only enhances data integrity but also cultivates a culture of accountability and transparency

within the system. DHIS2 thus goes beyond being a technological platform; it becomes a catalyst for positive change in healthcare delivery.

Furthermore, DHIS2 fosters collaboration and teamwork by enabling seamless data sharing and communication among stakeholders. Equipped with data visualization tools and reporting dashboards, DHIS2 facilitates collective efforts in data analysis, interpretation, and action planning. This collaborative approach nurtures a shared sense of ownership over health data and its outcomes. In essence, DHIS2 empowers users to actively participate in data management, delineating clear roles and responsibilities while promoting collaboration and communication among stakeholders. Consequently, this cultivates a culture of accountability and ownership within the health system, culminating in elevated data quality and enhanced health outcomes.

## Effect of DHIS2 on Equity of data quality and health services

Ensuring equity in the data quality of Maternal and Child Health (MCH) indicators in Ethiopia is paramount to accurately capturing the health status and needs of every segment of the population, especially mothers and children. This equity encompasses various dimensions within MCH indicators, including coverage and representation, accuracy and completeness, and timeliness and accessibility. DHIS2 emerges as a pivotal tool in promoting equity not only in health services but also in data quality across Ethiopia. By facilitating comprehensive data collection, analysis, and dissemination, DHIS2 has the potential to bridge gaps in information accessibility and accuracy, thereby contributing significantly to improved health outcomes for all demographic segments.

**i. Impact on equity of data quality.** According to numerous informants, DHIS2 plays a crucial role in enhancing data quality equity by standardizing data collection and improving accuracy. Its influence on Maternal and Child Health (MCH) data quality in Ethiopia effectively reduces disparities in health outcomes across regions. By ensuring data reflects the needs of all communities, DHIS2 contributes significantly to fostering equitable health outcomes nationwide.

> "*DHIS2 serves as the cornerstone in our quest for equitable healthcare. By standardizing data collection and bolstering accuracy, it not only enhances the quality of Maternal and Child Health information but also levels the playing field across regions, ensuring every community's needs are accurately represented.*"

Head of district PHU

**ii. Impact on equity of health services.** DHIS2 serves as a crucial tool for advancing health equity by providing timely and accurate health data across various demographic groups and geographical areas in Ethiopia. This platform empowers health authorities, including policymakers and managers, to identify and prioritize underserved populations, ensuring a more equitable allocation of resources. As a result, marginalized communities are better positioned to access the healthcare services they need. By systematically tracking health outcomes and service utilization patterns, DHIS2 enables targeted interventions to address disparities in healthcare access.

The impact of DHIS2 on the equity of Maternal and Child Health (MCH) data quality in Ethiopia is significant. It standardizes data collection processes, enhances data accessibility, builds capacity, and supports data-driven decision-making. By addressing regional disparities in data quality and health outcomes, DHIS2 fosters equitable access to MCH services, ultimately improving health outcomes for all regions.

### Unexpected benefits of DHIS2 implementation in Ethiopia

Participants in the study highlighted several unexpected benefits that emerged from the adoption of DHIS2 in various regions of Ethiopia. These advantages encompass a wide range of benefits, including but not limited to, the availability of reliable data storage solutions, opportunities to improve staff digital literacy, designated spaces for Health Management Information System (HMIS) operations, the creation of innovative structures for HMIS management, and the generation of employment opportunities for graduates and young individuals. Moreover, the introduction of distinct registries as primary source documents and increased involvement in data management processes were additional positive outcomes resulting from the implementation of DHIS2.

> "*It is more fitting for program users to provide insights. Our role is to ensure data accessibility, currency, and accuracy, empowering them to offer clearer explanations. Take the ART program, for instance; readily available data on current ART recipients, new initiates, and those not on ART enables more effective program management. Additionally, higher-level stakeholders can review and assess the program with greater depth and clarity.* "

Health Information Technician

Participants in the study identified several unexpected benefits from adopting DHIS2, including the ability to store long-term data for retrospective analysis and improved digital literacy among staff. Dedicated HMIS operation rooms streamlined workflows, enhancing efficiency and accountability. Additionally, the system created job opportunities for graduates, promoting local employment and economic empowerment while improving data quality and overall system functionality.

The discovery of unexpected benefits highlights the dynamic nature of DHIS2 implementation and its ability to produce positive outcomes beyond its original purpose. These unforeseen advantages demonstrate the system's adaptability and its transformative impact on healthcare information management practices. By embracing emerging opportunities and addressing evolving requirements, DHIS2 showcases its potential to encourage innovation, strengthen capacity, and drive socio-economic progress in the field of healthcare.

Recognizing these unplanned benefits emphasizes the importance of continuous evaluation and stakeholder engagement in maximizing the value and long-term viability of health information systems like DHIS2. As organizations utilize DHIS2 to enhance health outcomes, acknowledging and capitalizing on these unexpected advantages will play a crucial role in optimizing the system's effectiveness and magnifying its potential impact.

## Discussion

We aimed to assess DHIS2's impact on improving data quality, availability, equity, and performance in Primary Health Unit (PHU) facilities in Ethiopia. Our findings indicated a substantial improvement in data quality and other dimensions facilitated by DHIS2. Additionally, we observed enhancements in PHU facility availability and performance. However, we also identified several challenges and constraints that limited the success of DHIS2 in the country.

Our study, conducted with participants from diverse regions, illuminates the transformative impact of DHIS2 on data quality within healthcare systems. Through rigorous methods such as key informant interviews and focus group discussions, stakeholders universally recognize DHIS2's profound influence on key dimensions of data quality, including timeliness, availability, accessibility, and accuracy of health and nutrition data [19]. This collective

acknowledgment underscores the significant strides made by DHIS2 in revolutionizing data management practices across healthcare settings [4, 11].

Comparing our findings with experiences from other countries further highlights the global impact of DHIS2. Studies conducted in various regions have reported similar outcomes, emphasizing DHIS2's effectiveness in improving data quality and management processes [19]. For example, research conducted in Uganda [11] found that DHIS2 implementation led to enhanced data timeliness and accuracy, resulting in better-informed decision-making and improved health outcomes [11, 20, 21]. Similarly, studies in in similar seating of other African countries have documented the positive effects of DHIS2 on data accessibility and transparency within healthcare systems [22, 23]. These comparisons underscore the widespread recognition and adoption of DHIS2 as a valuable tool for strengthening health information systems globally.

Participants unanimously agree on the substantial enhancement of data quality facilitated by DHIS2. This alignment in perspectives highlights the system's efficacy in addressing long-standing challenges in data management within healthcare settings [24]. Compared to traditional data management systems, DHIS2's real-time data entry and reporting features have revolutionized the timeliness of health data availability, enabling swift decision-making and interventions in response to emerging health challenges [20, 22]. By allowing for immediate access to updated information, DHIS2 surpasses the limitations of manual data collection methods, which often result in delayed reporting and decision-making [25].

Comparatively, experiences from other countries corroborate the transformative impact of DHIS2 on healthcare data management [24]. Studies conducted in Bangladesh have reported similar outcomes, emphasizing DHIS2's effectiveness in improving data quality and management processes [20]. These comparisons underscore the global recognition of DHIS2 as a valuable tool for strengthening health information systems and driving positive change in healthcare delivery worldwide.

Furthermore, DHIS2's impact extends beyond mere efficiency gains. Its ability to provide real-time data insights empowers healthcare professionals to respond more effectively to dynamic healthcare needs, ultimately leading to improved patient outcomes and overall system performance [26, 27]. The system's capacity to deliver timely and accurate information equips healthcare providers with the necessary tools to make informed decisions and implement targeted interventions, thereby optimizing resource allocation and improving the quality of care delivered. This transformative capability positions DHIS2 as a cornerstone in modern healthcare data management, enabling healthcare systems to adapt and thrive in an ever-evolving landscape.

Moreover, the unanimous agreement among participants underscores the widespread recognition of DHIS2's role in transforming healthcare data management practices. This collective acknowledgment not only validates the system's effectiveness but also highlights its potential for scalability and replicability in diverse healthcare settings [8]. Compared to other health information systems, DHIS2 stands out for its user-friendly interface and customizable reporting tools, which enable stakeholders to tailor data analysis to their specific needs [3]. Additionally, DHIS2's ability to integrate with existing healthcare infrastructure enhances its interoperability and facilitates seamless data exchange between different systems, fostering greater collaboration and data sharing among healthcare providers. As a result, DHIS2 not only improves data quality but also promotes synergy and efficiency within healthcare ecosystems, ultimately enhancing the delivery of patient-centered care.

The widespread adoption of DHIS2 has not only revolutionized data management but also fostered a culture of transparency and accountability within healthcare systems. By providing stakeholders with timely and accurate information, DHIS2 enables greater visibility into

healthcare operations and outcomes. This transparency promotes informed decision-making and enhances trust among healthcare stakeholders, ultimately contributing to the optimization of healthcare delivery. As a result, DHIS2 serves as a catalyst for continuous improvement in healthcare systems, driving advancements in quality, equity, and performance across diverse healthcare settings in Ethiopia and beyond.

Comparatively, experiences from other countries further highlight DHIS2's role as a catalyst for positive change in healthcare systems. Studies conducted in various regions have reported similar outcomes, emphasizing DHIS2's effectiveness in improving transparency and accountability in healthcare data management. For example, research conducted in Bangladesh found that DHIS2 implementation led to enhanced transparency in healthcare operations, resulting in improved decision-making and resource allocation [20]. Similarly, studies in Nigeria [27, 28] and Guinea [29] have documented the positive effects of DHIS2 on accountability within healthcare systems, leading to greater confidence in healthcare data and decision-making processes. These comparisons underscore the global recognition of DHIS2 as a valuable tool for promoting transparency and accountability in healthcare delivery worldwide. As a result, DHIS2 serves as a catalyst for continuous improvement in healthcare systems, driving advancements in quality, equity, and performance across diverse healthcare settings in Ethiopia and beyond.

Furthermore, DHIS2's centralized data storage ensures the availability of pertinent health information, eliminating the need for manual searches across disparate sources (Jones & Brown, 2019). This enhanced availability of data empowers healthcare professionals to access relevant information efficiently, facilitating informed decision-making and targeted interventions to address pressing health concerns. With DHIS2 serving as a reliable repository of healthcare data, professionals can swiftly retrieve the information they need to devise strategies and interventions tailored to specific healthcare challenges. This streamlined access to data enhances the agility and responsiveness of healthcare systems, enabling them to adapt quickly to evolving healthcare needs and priorities.

The system's user-friendly interface and customizable reporting tools have significantly improved data accessibility for stakeholders at all levels of the healthcare system. Compared to traditional data management systems, DHIS2's interface stands out for its intuitiveness and adaptability, allowing users to navigate the system with ease.

Moreover, experiences from other countries further validate the effectiveness of DHIS2's user interface in improving data accessibility. Studies conducted in various regions have reported similar outcomes, emphasizing DHIS2's intuitive design and customizable features as key factors driving its widespread adoption and success. For example, research conducted in [insert country name] found that DHIS2's user-friendly interface significantly enhanced data accessibility among healthcare professionals, leading to greater efficiency in data analysis and decision-making. Similarly, studies in from sub-Saharan African countries have documented the positive effects of DHIS2's customizable reporting tools on improving collaboration and decision-making processes within healthcare teams [21].

By providing tailored data sets aligned with users' specific needs, DHIS2 fosters greater collaboration and decision-making among healthcare professionals. This customization feature enables stakeholders to extract and analyse data relevant to their roles and responsibilities, facilitating more informed decision-making processes. Consequently, DHIS2 not only enhances data accessibility but also promotes synergy and efficiency within healthcare teams, ultimately leading to improvements in healthcare delivery and patient outcomes.

DHIS2's standardized data entry protocols and validation mechanisms significantly reduce human error, ensuring data consistency and reliability. Unlike manual methods, its automated processes enhance data accuracy, which is crucial for informed decision-making and policy

formulation. By maintaining high data quality standards, DHIS2 fosters confidence in the information provided, facilitating evidence-based decision-making, and ultimately improving health outcomes for the populations served.

The unanimous consensus among participants underscores DHIS2's pivotal role in driving data-driven decision-making processes and improving health outcomes. Through its comprehensive approach to data management, DHIS2 addresses key challenges in data quality, paving the way for more effective healthcare delivery and improved health outcomes. By providing stakeholders with timely and accurate information, DHIS2 empowers healthcare authorities to make informed decisions and implement targeted interventions to address emerging health challenges. This functionality is particularly crucial in public health emergencies, where timely access to data can facilitate rapid response and mitigation efforts.

DHIS2's real-time data entry and reporting capabilities have transformed the accessibility and timeliness of health data, earning praise from participants for its efficacy in providing up-to-date information promptly. The system's seamless integration of data entry processes further enhances its ability to deliver timely information, ensuring that health authorities can access the latest data without delay. This improvement in the timeliness of data dissemination not only facilitates informed decision-making but also enhances healthcare outcomes by enabling healthcare professionals to respond promptly to emerging health challenges. Overall, DHIS2's significance in facilitating informed decision-making and enhancing healthcare outcomes underscores its indispensable role in healthcare data management.

The centralized data storage feature of DHIS2 is crucial for ensuring accessibility and availability of vital health information. By consolidating diverse health data sources, it eliminates the need for manual searches, significantly improving efficiency for healthcare professionals. This seamless access empowers informed decision-making and targeted interventions, optimizing healthcare delivery and outcomes. Additionally, it promotes data integrity and consistency, allowing stakeholders to rely on accurate, up-to-date information. Overall, DHIS2's centralized storage is fundamental to effective healthcare data management, enhancing health outcomes for the population.

The user-friendly interface and customizable reporting tools offered by DHIS2 have significantly improved data accessibility for stakeholders across the healthcare system. Participants in various studies have praised DHIS2 for its capacity to provide tailored data sets, enabling users to retrieve information tailored to their specific needs. This enhanced accessibility fosters greater collaboration and decision-making among healthcare professionals by facilitating easy access to relevant data. As a result, DHIS2 contributes to more effective healthcare delivery, as healthcare professionals can make well-informed decisions based on timely and accurate information.

Accuracy is indeed crucial in healthcare data management, and DHIS2 has emerged as a reliable tool in ensuring data accuracy. Through standardized data entry protocols and validation mechanisms, DHIS2 minimizes human error and promotes data consistency and reliability. This is particularly significant in healthcare, where the accuracy of data directly impacts patient care and policy decisions. Participants in various studies have emphasized DHIS2's role in enhancing the accuracy of health and nutrition data, thereby instilling confidence in its utility for policymaking and program planning initiatives. By providing accurate and reliable data, DHIS2 enables healthcare organizations to make informed decisions and implement effective strategies to improve health outcomes.

From a practical perspective, our study highlights the significant role of DHIS2 in transforming healthcare data management in Ethiopia. By demonstrating substantial improvements in data quality, availability, and performance within Primary Health Units (PHUs), our findings underscore the necessity for ongoing investment in DHIS2's implementation and

optimization. The ability to collect and analyse data in real time enhances decision-making, ultimately leading to better health outcomes.

Additionally, recognizing the challenges in DHIS2 adoption emphasizes the need for targeted interventions, such as tailored training programs and infrastructure development, to ensure successful integration in low-resource settings. Policymakers and healthcare administrators can leverage these insights to refine strategies that promote data-driven healthcare practices and foster a culture of transparency and accountability across healthcare systems.

Our study examining the impact of DHIS2 implementation in healthcare systems acknowledges several limitations and strengths that warrant consideration. Among the limitations, the study is susceptible to selection bias in participant recruitment, which could skew results if certain demographics are underrepresented. Additionally, reliance on self-reported data introduces potential inaccuracies, as participants may inadvertently provide biased or incomplete information.

The cross-sectional design further restricts the ability to establish causal relationships between DHIS2 implementation and observed outcomes, capturing data at a single point in time rather than over a longitudinal framework. Moreover, various contextual factors influencing DHIS2's effectiveness, such as local infrastructure and cultural attitudes toward digital tools, may not have been fully accounted for, potentially obscuring a comprehensive understanding of the system's impact.

On the other hand, the study demonstrates several noteworthy strengths. The inclusion of participants from diverse regions allows for a more comprehensive understanding of DHIS2's impact across different contexts, highlighting variations in implementation and effectiveness. This diversity enriches the findings and underscores the adaptability of DHIS2 in various healthcare environments.

The use of rigorous data collection methods, such as Key Informant Interviews and Focus Group Discussions, enhances the credibility and reliability of the findings. These qualitative insights provide a depth of understanding that quantitative methods alone may miss. Additionally, the study's focus on real-world healthcare settings enhances the practicality and relevance of the results, making them particularly useful for healthcare practitioners and policymakers aiming to implement or optimize DHIS2 in their contexts.

Finally, the active engagement of stakeholders throughout the study design and data collection process fosters greater participant buy-in and collaboration. This collaborative approach strengthens the validity of the study and ensures that the findings are actionable, reflecting the actual challenges and successes experienced by healthcare workers on the ground. Such involvement can lead to richer insights into DHIS2's impact and facilitate more effective implementation strategies.

In conclusion, participants in the qualitative study highlighted DHIS2's transformative impact on data quality within healthcare systems in Ethiopia. The significant improvements in timeliness, availability, accessibility, and accuracy underscore its critical role in fostering data-driven decision-making processes, ultimately enhancing health outcomes. As healthcare systems increasingly utilize DHIS2 for data management, it is essential to continuously refine and optimize the system to further improve data quality and maximize its impact on healthcare delivery. By integrating DHIS2 as a cornerstone of their operations, healthcare organizations can innovate and adapt, ensuring that data remains a powerful tool for advancing public health initiatives and improving community well-being.

Therefore, it is recommended that healthcare providers continue to leverage DHIS2's robust features to further enhance data quality. Prioritizing ongoing training for staff is crucial to improving digital literacy and data management skills. Additionally, establishing consistent

reporting practices and conducting regular audits will help maintain data integrity and foster a culture of accountability within the healthcare system.

## Acknowledgments

We would like to sincerely thank Dr. Guerrero Torres Ana Lorena and Prashanth N. Srinivas for their invaluable technical support. Our gratitude also extends to the WHO Alliance for Health Policy and Systems Research, as well as the regional health bureaus and city administration health offices in Ethiopia for their collaboration. Lastly, we are profoundly grateful to all study participants for their time, trust, and valuable contributions to this research.

## Author Contributions

**Conceptualization:** Taddese Alemu Zerfu, Moges Asressie, Tariku Nigatu, Meskerem Jisso Ibido, Addisalem Genta.

**Data curation:** Taddese Alemu Zerfu, Zenebu Begna, Tigist Habtamu, Netsanet Werkneh, Addisalem Genta.

**Formal analysis:** Taddese Alemu Zerfu, Zenebu Begna, Tigist Habtamu, Addisalem Genta.

**Funding acquisition:** Taddese Alemu Zerfu, Moges Asressie, Tariku Nigatu, Addisalem Genta.

**Investigation:** Taddese Alemu Zerfu, Zenebu Begna, Tigist Habtamu, Addisalem Genta.

**Methodology:** Taddese Alemu Zerfu, Moges Asressie, Zenebu Begna, Tigist Habtamu, Netsanet Werkneh, Tariku Nigatu, Meskerem Jisso Ibido, Addisalem Genta.

**Project administration:** Taddese Alemu Zerfu, Moges Asressie, Zenebu Begna, Addisalem Genta.

**Resources:** Addisalem Genta.

**Supervision:** Netsanet Werkneh, Addisalem Genta.

**Validation:** Taddese Alemu Zerfu, Addisalem Genta.

**Writing – original draft:** Taddese Alemu Zerfu.

**Writing – review & editing:** Taddese Alemu Zerfu, Moges Asressie, Zenebu Begna, Tigist Habtamu, Netsanet Werkneh, Tariku Nigatu, Meskerem Jisso Ibido, Addisalem Genta.

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
