## [Decision Letter · Decision Letter 0]

20 Sep 2024

PONE-D-24-29622Unveiling DHIS2's Impact on Enhancing Data Quality, Availability, and Performance of PHU Facilities: Insights from EthiopiaPLOS ONE

Dear Dr. Zerfu,

Thank you for submitting your manuscript to PLOS ONE. After careful consideration, we feel that it has merit but does not fully meet PLOS ONE’s publication criteria as it currently stands. Therefore, we invite you to submit a revised version of the manuscript that addresses the points raised during the review process.

We look forward to receiving your revised manuscript.

Kind regards,

Addisalem Workie Demsash

Academic Editor

PLOS ONE

Journal Requirements:

2. In the ethics statement in the Methods, you have specified that verbal consent was obtained. Please provide additional details regarding how this consent was documented and witnessed, and state whether this was approved by the IRB

3. Thank you for stating the following financial disclosure: Funding for this project was obtained from WHO Alliance for Health Policy and Systems Research.  

4. Thank you for stating the following in the Acknowledgments Section of your manuscript: We extend sincere appreciation to the WHO Alliance for Health Policy and Systems Research

and Dr Prashanth N Srinivas for their invaluable technical and financial support to this study.

Please remove any funding-related text from the manuscript and let us know how you would like to update your Funding Statement. Currently, your Funding Statement reads as follows: Funding for this project was obtained from WHO Alliance for Health Policy and Systems Research.

5. In the online submission form, you indicated that the data supporting the findings presented in the study can be requested from the corresponding author at tadalzerfu@gmail.com.

Reviewers' comments:

Reviewer's Responses to Questions

**Comments to the Author**

1. Is the manuscript technically sound, and do the data support the conclusions?

Reviewer #1: Yes

Reviewer #2: Yes

2. Has the statistical analysis been performed appropriately and rigorously? 

Reviewer #1: No

Reviewer #2: Yes

3. Have the authors made all data underlying the findings in their manuscript fully available?

Reviewer #1: No

Reviewer #2: Yes

4. Is the manuscript presented in an intelligible fashion and written in standard English?

Reviewer #1: Yes

Reviewer #2: Yes

5. Review Comments to the Author

Reviewer #1: Review Report

The manuscript titled " Unveiling DHIS2's Impact on Enhancing Data Quality, Availability, and Performance of PHU Facilities: Insights from Ethiopia" investigates the impact influencing the data quality, availability and performance in Ethiopia. The study is timely and relevant, addressing the increasing importance of digital health technologies in resource-limited settings. The manuscript addresses an important and timely topic with a well-structured study design and robust frameworks. However, to enhance the clarity, depth, and impact of the study, the authors should consider the revisions and recommendations outlined below. This will help ensure the findings are both comprehensible and actionable for a broader audience.

Strengths

The topic is highly relevant, given the global push towards digital health solutions, particularly in low-resource settings. The use of the qualitative studies by selecting four cities including Addis Ababa it sounds a robust theoretical framework for understanding the DHIS2's Impact on Enhancing Data Quality, Availability, and Performance of PHU Facilities. The study employs a well-defined sampling technique and uses ATLAS.ti software for data analysis. Ethical approval and participant consent were obtained, ensuring the study adheres to ethical research standards.

see the attached supporting file

Reviewer #2: 1. The problem is clearly mentioned

2. The sampling procedure, convenience and purposive sampling strategies were utilized to identify and recruit participants for key informant interviews and focus group discussions. Explain separate for which convenience sampling? KII or FGD and purposive sampling?

3. In sample size and Sampling Technique section said that a total of 30 KIIs and 16 FGDs were conducted. Not clear, what does it mean how money participants were participate in 16 FGD?

4. the result and discussion part is vast and bulky to read and understand so focus om the key point

6. PLOS authors have the option to publish the peer review history of their article (what does this mean?). If published, this will include your full peer review and any attached files.

Reviewer #1: No

Reviewer #2: No

---

## [Author Response · Author response to Decision Letter 0]

8 Oct 2024

Point by Point Response

Our data has been uploaded to a public repository and is accessible via a unique digital object identifier (DOI): [doi:10.5061/dryad.k98sf7mh2](https://doi.org/10.5061/dryad.k98sf7mh2).

Response to reviewer – 1 

1) The manuscript titled " Unveiling DHIS2's Impact on Enhancing Data Quality, Availability, and Performance of PHU Facilities: Insights from Ethiopia" investigates the impact influencing the data quality, availability and performance in Ethiopia. The study is timely and relevant, addressing the increasing importance of digital health technologies in resource-limited settings. The manuscript addresses an important and timely topic with a well-structured study design and robust frameworks. However, to enhance the clarity, depth, and impact of the study, the authors should consider the revisions and recommendations outlined below. This will help ensure the findings are both comprehensible and actionable for a broader audience. 

The topic is highly relevant, given the global push towards digital health solutions, particularly in low-resource settings. The use of the qualitative studies by selecting four cities including Addis Ababa it sounds a robust theoretical framework for understanding the DHIS2's Impact on Enhancing Data Quality, Availability, and Performance of PHU Facilities. The study employs a well-defined sampling technique and uses ATLAS.ti software for data analysis. Ethical approval and participant consent were obtained, ensuring the study adheres to ethical research standards.

Response - 1

General response

Thank you for your thoughtful feedback. We appreciated your recognition of the study's relevance and robust design. We carefully considered your revisions and recommendations to enhance clarity and impact, ensuring our findings were comprehensible and actionable for a broader audience. Your insights on our sampling technique and ethical standards were also valued, and we made improvements to the manuscript accordingly.

2) The study lacks practical implications, and the author should be including the strength and limitations that ware faced in the provided study.

Response - 2

Thank you, this is valuable feedback! We acknowledged that the study lacked practical implications and recognized the importance of including the strengths and limitations faced during the research. We addressed these points in the revised manuscript to provide a more comprehensive understanding of the study's contributions and challenges.

3) There are some areas where the language could be clearer and more concise. Review the manuscript for clarity and conciseness. Ensure that technical terms are well-defined and that the narrative flow is logical and easy to follow.

Response – 3

We recognized that some areas of the manuscript needed clearer and more concise language. We then reviewed the text thoroughly to enhance clarity, ensuring that technical terms were well-defined and that the narrative flow was logical and easy to follow. Your suggestions helped improve the overall readability of the manuscript.

4) The practical implications of the findings could be more explicitly stated. The authors should emphasize how the findings can inform the impact on enhancing data quality, availability, and performance DHIS2 systems in Ethiopia. Provide specific recommendations for healthcare providers, policymakers, and technology developers.

Response – 4 

We acknowledged that the practical implications of our findings needed explicitly. In the revised manuscript, we emphasized how our findings inform the enhancement of data quality, availability, and performance of DHIS2 systems in Ethiopia. We also provided specific recommendations for healthcare providers, policymakers, and technology developers to ensure our insights are actionable and relevant.

5) Comments to the Abstract - Ensure the abstract succinctly summarizes the key findings and their significance. Consider including specific recommendations derived from the study.

Response – 5

Thank you for your feedback. We have now refined the results section to succinctly summarize the key findings and included specific recommendations derived from the study, enhancing clarity and applicability. The revised results section reads:

“Participants from various regions reported significant enhancements in the timeliness, completeness, accuracy, and accessibility of health data following the implementation of DHIS2. While some concerns were raised regarding variations in reporting intervals, the consensus indicated marked improvements in data management processes. DHIS2 standardized data collection methods, enabling healthcare providers to input and access data in real-time. This advancement fostered greater accountability and transparency within the healthcare system. Additionally, unexpected benefits arose, including increased digital literacy among staff, equipping them with necessary skills for effective data management, and the creation of job opportunities, particularly for youth. Ultimately, DHIS2 emerged as a pivotal tool for enhancing data quality and promoting health service equity across Ethiopia.”

The conclusion and recommendations now read:

“DHIS2 significantly improved data quality and accessibility in Ethiopia, enhancing healthcare management and fostering accountability across various health facilities. It is recommended that healthcare providers continue to leverage DHIS2’s robust features to further enhance data quality. Prioritizing ongoing training for staff is essential to improve digital literacy and data management skills. Additionally, establishing consistent reporting practices and conducting regular audits will help maintain data integrity and cultivate a culture of accountability within the healthcare system.”

6) Comments on the Introduction - Provide more context on the DHIS2's Impact on Enhancing Data Quality, Availability, and Performance of PHU Facilities in Ethiopia. Highlight the gap in knowledge regarding DHIS2 adoption in low-resource settings.

Response – 6 

Thank you for your valuable feedback on the introduction. We have expanded the section to provide more context on DHIS2's impact on enhancing data quality, availability, and performance within primary healthcare units (PHUs) in Ethiopia. Additionally, we highlighted the existing gaps in knowledge regarding DHIS2 adoption in low-resource settings, specifically focusing on contextual factors that influence successful integration, such as infrastructure, training needs, and cultural attitudes.

7) Comments on the Literature Review - Expand the literature review to include more studies on DHIS2 related to data quality, availability, and performance globally and in similar contexts. This will help position the study within the broader research landscape.

Response – 7

Thanks, we have conducted a thorough review of relevant studies and integrated key findings to enhance the depth and relevance of our literature review.

8) Comments on the Results session - Should be write clearly presented with appropriate saturated levels.

Response – 8

Thank you for your feedback regarding the Results section. We have revised the content to ensure that the findings are clearly presented and that the saturation levels are appropriately highlighted. Your input has been instrumental in enhancing the clarity and comprehensibility of this section. We appreciate your guidance in improving the overall quality of our manuscript.

9) Comments on the Discussion session - Discussion should be rewrite succinctly. To make suitable and make easy for comments towards the reviewer the author should use continuous line number for all your manuscript.

Response – 9

Thank you for your feedback. We revised this part to enhance clarity and focus. Additionally, we implemented continuous line numbering throughout the manuscript to facilitate easier referencing for reviewers. Your insights were invaluable in improving our work.

Response to reviewer – 2

1. I am positive this paper is published 

2. The problem is clearly mentioned 

Response – 1 

Thank you for your positive feedback! We appreciate your acknowledgment of the clarity in stating the problem in the paper. 

3. The sampling procedure, convenience and purposive sampling strategies were utilized to identify and recruit participants for key informant interviews and focus group discussions. Explain separate for which convenience sampling? KII or FGD and purposive sampling?

Response – 2 

We used convenience sampling for focus group discussions (FGDs) to recruit participants who were readily available, enabling us to efficiently gather diverse perspectives. In contrast, we employed purposive sampling for key informant interviews (KIIs) to select individuals with specific expertise, ensuring we obtained in-depth insights that aligned with our research objectives. 

Now reading as: ‘’ … Convenience and purposive sampling strategies were utilized to identify and recruit participants for focus group discussions and key informant interviews, respectively.’’

4. In sample size and Sampling Technique section said that a total of 30 KIIs and 16 FGDs were conducted. Not clear, what does it mean how money participants were participate in 16 FGD?

Response – 3 

Thank you for asking about this. Each focus group typically consisted of 8 participants, although a few had between 7 and 9 participants. This resulted in approximately 132 participants across the 16 FGDs. This approach allowed us to gather a wide range of perspectives and enrich the discussions. 

Now reading as: ‘’ … Purposive sampling techniques were employed to select participants based on their roles within primary healthcare institutions, level of training, and professional experience. This method ensured the inclusion of a diverse array of stakeholders, from health managers at district hospitals to performance monitoring team leaders at health facilities. In total, we conducted 30 KIIs and 16 FGDs. Most focus groups consisted of 8 participants, although a few included between 7 and 9 participants, resulting in approximately 132 participants across the 16 FGDs.’’

---

## [Decision Letter · Decision Letter 1]

12 Nov 2024

Unveiling the Role of DHIS2 in Enhancing Data Quality and Accessibility in Primary Healthcare Facilities: Evidence from Ethiopia

PONE-D-24-29622R1

Dear Dr. Zerfu,

We’re pleased to inform you that your manuscript has been judged scientifically suitable for publication and will be formally accepted for publication once it meets all outstanding technical requirements.

Kind regards,

Addisalem Workie Demsash

Academic Editor

PLOS ONE

---

## [Editor Report · Acceptance letter]

20 Nov 2024

PONE-D-24-29622R1 

PLOS ONE

Dear Dr. Zerfu, 

I'm pleased to inform you that your manuscript has been deemed suitable for publication in PLOS ONE. Congratulations! Your manuscript is now being handed over to our production team.

Kind regards, 

on behalf of

Mr. Addisalem Workie Demsash 

Academic Editor

PLOS ONE